# Relationship between Bioelectrical Impedance Phase Angle and Upper and Lower Limb Muscle Strength in Athletes from Several Sports: A Systematic Review with Meta-Analysis

**DOI:** 10.3390/sports11050107

**Published:** 2023-05-18

**Authors:** Everton Cirillo, Alberto Pompeo, Fabiane Tavares Cirillo, José Vilaça-Alves, Pablo Costa, Rodrigo Ramirez-Campillo, Antonio Carlos Dourado, José Afonso, Filipe Casanova

**Affiliations:** 1Centro de Investigação em Desporto, Educação Física, Exercício e Saúde (CIDEFES), Lusófona University, 1749-024 Lisboa, Portugal; evertoncirillo@gmail.com (E.C.); albertopompeo@hotmail.com (A.P.); 2Sports Science Department, State University of Londrina (UEL), Londrina 86057-970, Brazil; fabianetcirillo@gmail.com (F.T.C.); dourado99@gmail.com (A.C.D.); 3Department of Sport, Exercise and Health Sciences at the University of Trás-os-Montes and Alto Douro, 5000 Vila Real, Portugal; josevilaca@utad.pt; 4Centro de Investigação em Desporto, Saúde e Desenvolvimento Humano (CIDESD), 5000 Vila Real, Portugal; 5Exercise Physiology Laboratory, Department of Kinesiology, California State University, Fullerton, CA 92831, USA; pcosta@fullerton.edu; 6Exercise and Rehabilitation Sciences Institute, Faculty of Rehabilitation Sciences, Universidad Andres Bello, Santiago 7591538, Chile; rodrigo.ramirez@unab.cl; 7Centre for Research, Education, Innovation and Intervention in Sport (CIFI2D), Faculty of Sport of the University of Porto, 4200-450 Porto, Portugal

**Keywords:** sport, performance, physical-physiological assessment, body composition

## Abstract

The phase angle (PhA) of bioelectrical impedance is determined by primary factors such as age, body mass index and sex. The researchers’ interest in applying PhA to better understand the skeletal muscle property and ability has grown, but the results are still heterogeneous. This systematic review with a meta-analysis aimed to examine the existence of the relationship between PhA and muscle strength in athletes. The data sources used were PubMed, Scielo, Scopus, SPORTDiscus, and Web of Science and the study eligibility criteria were based on the PECOS. The searches identified 846 titles. From those, thirteen articles were eligible. Results showed a positive correlation between PhA and lower limb strength (r = 0.691 [95% CI 0.249 to 0.895]; *p* = 0.005), while no meta-analysis was possible for the relationships between PhA and lower limb strength. Furthermore, GRADE shows very low certainty of evidence. In conclusion, it was found that most studies showed a positive correlation between PhA and vertical jump or handgrip strength. The meta-analysis showed the relationship between PhA and vertical jump, however, little is known for the upper limbs as was not possible to perform a meta-analysis, and for the lower limbs we performed it with four studies and only with vertical jump.

## 1. Introduction

During the past three decades, the bioelectrical impedance test (BIA) has been widely used by the research community to extend knowledge about body composition [1,2]. The BIA offers a simple approach to identifying biomarkers of cell damage and cell death in many populations (e.g., athletes, elderly, college students), determining the quality of the cell membrane throughout the body, and representing fluid distribution in individuals [1,2]. Albeit the body and its conductivity are not a uniform cylinder and constant, respectively, some relationship can be established between the impedance quotient (Length^2^/R [R stands for resistance]) and the volume of water, which contains electrolytes that conduct the electrical current through the body. Therefore, it is easier to measure height than conductive length, which is usually from wrist to ankle [3].

Moreover, due to the inherent field inhomogeneity in the body, the empirical relationship between lean body mass (typically 73% water) and height^2^/R describes an equivalent cylinder, which must be matched to real geometry by an appropriate coefficient [3]. The basic principle is based on the fact the human body is composed of a set of five “cylinders” (i.e., two arms, two legs and trunk) that offer different resistances to the passage of low-intensity electrical current and that stability of the tissue hydration [4]. Moreover, it is analyzed based on measurements of the total resistance of the body to the passage of electrical current of low-amplitude (800 mA) and high-frequency (50 kHz). For this purpose we have chosen to use the BIA, which measures several indicators such as resistance (R), reactance (Xc), and the phase angle (PhA), and it is also a reliable, low-cost and easy-to-use method [3].

The PhA is acquired through the relationship between the measurements of R and Xc, in which the electrical current passes through the body and is briefly stored in cell membranes. This storage causes a drop in the current voltage, creating a phase shift and resulting in an angle transforming the relationship between Xc and R, respectively [3]. These values (men—7.50° ± 0.60, women—6.71° ± 0.69) are proposed as an index of muscle fitness [5], with the relationship between PhA and cellular health increasing almost linearly [6]. Low values of PhA (18–19 years: men—6.82° ± 0.77; women—5.93° ± 0.69) are consistent with an inability of cells to store energy [7], and high values of PhA (14–18 years: men—8.01° ± 0.83; women—7.23° ± 0.89) are consistent with large amounts of intact cell membranes, reflecting the ratio of body cell mass to fat-free mass [8]. Factors such as age, body mass index and sex are primary determinants of PhA [7].

Muscle strength and power in the lower limbs are both important physiological characteristics in sports practitioners, especially in sports involving high-speed running and jumping, where their development is linked to high performance in high-level physical abilities [9]. These abilities are highly dependent on age, sex, morphological characteristics and level of physical fitness and must be considered during a test and the interpretation of its results [7,9,10,11,12]. To assess lower limb strength, vertical jump tests (e.g., countermovement jump—CMJ) and one repetition maximum (1RM), for example, can be used, especially where sprints or jumps are determinants of performance goals [8,13]. For the upper limbs, handgrip strength (HGS) provides an objective index of the integrity of upper limb functions, is a low-cost and easily applicable way to measure muscle strength, and has been established as a reliable clinical method associated with the general state of muscle strength [14].

Research interest in the application of PhA in athletes as an index of skeletal muscle properties, especially body water distribution in the whole body (WB) and/or limbs, has grown, but data delivered heterogeneous results. In healthy subjects, the PhA ranges from 5 degrees (°) to 7° and in well-trained athletes it may reach 8.5° [15]. The association of WB PhA with muscle performance in 117 adult athletes from different sports was evaluated in order to investigate whether regional PhA could be a better indicator of muscle performance compared to the WB while accounting for lean soft tissue (LST) [16]. The authors reported that PhA may have the potential to be used as a marker of functional muscle mass, which is important when it is intended to assess the muscular performance of athletes.

According to the pattern established for the searches of the present study, systematic reviews were found dealing with the connection between PhA in physical activities and sports and the evaluation of body composition, non-athletic children and adolescents, resistance training for older adults, muscle strength and aerobic fitness in different populations, oxidative stress, factors related to maturity in adolescent athletes and diseases (e.g., cardiovascular, cancer, and obesity). No reviews reported specifically the relationships between PhA with strength in athletes (lower and upper limbs), perhaps denoting the interest in the field still has not achieved critical mass. This demonstrates there is a gap in the literature in this domain, in which we expected that this research may contribute to the construction of knowledge and state of the art for the sport sciences. Therefore, the goal of this systematic review with meta-analysis was to assess the relationship between PhA (assessed through BIA) and the strength of lower and upper limbs of athletes of both sexes in different sports.

## 2. Materials and Methods

This review was registered on the OSF platform on 30 November 2021, one day before the searches were performed (project: https://osf.io/pmhgq/ (accessed on 21 April 2023); registration: https://osf.io/f5vxy (accessed on 21 April 2023)). PRISMA 2020 recommendations [17] (https://www.mdpi.com/article/10.3390/sports11050107/s1, Appendix A: PRISMA Checklist), and Cochrane’s guidelines [18] were followed.

### 2.1. Eligibility Criteria

We included original research published in peer-reviewed journals, with no date or language limit. Conference abstracts, even if published in peer-reviewed journals, were excluded. Eligibility criteria followed the PECOS approach: (i) participants were healthy athletes from any sport, regardless of sex or age; (ii) participants had exposure to regular sports training; (iii) comparators were optional; (iv) outcomes had to include PhA assessed through bioelectrical impedance and at least one of the following outcomes: lower limb strength (e.g., vertical jump, 1RM leg-press) and/or upper limb strength (e.g., medicine ball throwing, 1RM bench press); (v) no limitation was placed regarding study design.

### 2.2. Information Sources

Initial searches were performed on 1 December 2021, with updates on 30 September 2022, in the following databases without applying filters: Scielo, PubMed, Scopus, Web of Science and SPORTDiscus. Manual searches were also performed within the reference lists of the studies that were included. Subsequently, a snowballing citation tracking was performed on Web of Science. Then two external experts (Ph.D. holders who had published research on the topic in many publications on the Web of Science) were consulted to provide further suggestions for potentially relevant studies. Following the recommendations of Higgins [18], we searched errata and retractions of the included studies. In cases where there were pre-registered protocols and/or complementary files related to the included studies, these were also retrieved.

### 2.3. Search Strategy

The full search strategies for each database are presented in Table 1.

### 2.4. Selection Process

EC and JA independently screened each record retrieved. In cases of disagreements between the two authors, FC provided arbitration until consensus was achieved. Automated removal of duplicates was performed using EndNoteWeb (ClarivateTM), but further manual removal of duplicates was required.

### 2.5. Data Collection Process

AP and EC independently collected data from the reports. In cases of disagreements between two authors, FC provided arbitration until consensus was achieved. In cases where relevant data was missing and/or additional details were required, the authors of the studies were contacted by e-mail and ResearchGate, and the required information was solicited. In cases with no response, the studies were only excluded if the missing data were directly linked to the eligibility criteria. No automation tools were used.

### 2.6. Data Items

Outcomes: PhA assessed through bioelectrical impedance and at least one of the following outcomes: lower limb strength (vertical jump, 1RM leg-press) and/or upper limb strength (medicine ball throwing, 1RM bench press).

Additional variables: participant-related characteristics (sample size, age, sex, competitive level, sport, sport type, exposure to regular sports training), assessment-related features (specific assessments that were performed, number and blinding of testers, familiarization with testing procedures, time of season during the assessments, reliability of assessments), and other study-related information (country, funding, competing interests).

### 2.7. Study Risk of Bias Assessment

Risk of bias was performed at the outcome level using Cochrane’s RoBANS [19], with a worst-case scenario provided for the study level. Six domains were assessed: selection of participants, confounding variables, measurement of intervention (exposure), blinding of outcome assessment, incomplete outcome data and selective outcome reporting. EC and JA independently assessed the risk of bias. In cases with disagreements between the two authors, FC provided arbitration until consensus was achieved.

### 2.8. Effect Measures

For the main outcomes, means and standard deviations were presented (or median and interquartile range when appropriate). Correlation values between PhA and strength tests were presented alongside their confidence intervals.

### 2.9. Synthesis Methods

We established a minimum of three studies that provided data (e.g., correlation) for the same outcome [20,21] to avoid small sample sizes [22,23]. The main meta-analysis was based on correlation coefficients (r) as a well-known effect size estimate. Correlational data was used to compute the effect size (and its variance) for each study. For correlations, all computations were carried out using Fisher’s Z-transformed values. Correlation coefficients were entered along with the corresponding sample size or related data (i.e., standard error; variance; Fisher’s Z; t value; *p*-value), and the software (Comprehensive Meta-Analysis program, version 2; Biostat, Englewood, NJ, USA) was set to produce pooled r or with 95% confidence interval (CI) using random effects models. The inverse variance random effects model for meta-analyses was used because it allocates a proportionate weight to trials based on the size of their individual standard errors [24] to better account for inaccuracy in the estimation of between-study variance [25] and enables analysis while accounting for heterogeneity across studies [26]. The pooled effect size for r was classified as small (≤0.1), moderate (0.1–0.29) or large (≥0.30) [27].

The impact of study heterogeneity was assessed using the *I*^2^ statistic, with values of <25%, 25–75%, and >75% representing low, moderate and high levels of heterogeneity, respectively [28]. Computation of meta-regression was planned with at least 10 studies per covariate [18].

### 2.10. Risk of Reporting Bias Assessment

Risk of reporting bias was not assessed as the minimum number of 10 studies per comparison was not achieved. Planned assessments can be consulted in the registered protocol.

### 2.11. Certainty Assessment

Certainty or confidence in the body of evidence for each outcome was assessed using GRADE [29], considering five dimensions: risk of bias in studies, indirectness, inconsistency, imprecision and risk of publication bias [30,31].

## 3. Results

### 3.1. Study Selection

The initial searches (12/01/2021), plus update, (30/09/2022) retrieved 846 (786 + 60) records: PubMed: (56 + 21 = 77); Scielo: (1 + 0 = 1); Scopus: (545 + 0 = 545); SPORTDiscus: (83 + 6 = 89); and Web of Science: (101 + 33 = 134), of which 260 were duplicates. Of the 586 records screened, 158 were excluded due to not being empirical studies in peer-reviewed journals, and 407 were excluded due to not fulfilling one or more PECOS criteria. Four records fulfilled PECOS criteria, but were published in conference proceedings, not in peer-reviewed journals [32,33,34,35]. Seventeen studies were deemed eligible for full text analysis [9,11,15,16,36,37,38,39,40,41,42,43,44,45,46,47,48]. Of these, two studies were excluded since PhA was not assessed [39,46] and one because strength was not assessed [36]. Two studies assessed PhA and strength but had no correlation data reported, which was required to fulfil eligibility criteria. All the authors of these studies were contacted through email and, when available, through ResearchGate. The authors of two studies provided the necessary data [40,47], while no response was obtained for the other two studies [42,44], which were therefore excluded. Therefore, ten studies were included after the database searches [9,11,16,37,38,40,41,43,45,47]. The reference lists of these studies were then searched to retrieve potentially relevant titles that had not emerged in our initial searches. Two titles were promising but were excluded as they did not assess PhA [48,49].

Afterwards, searches were carried out in the reference lists of the ten articles, where four studies were found, of which two were removed because they were duplicates and two were included [50,51]. We also performed a snowballing citation tracking for the 12 included studies in Web of Science on 20 December 2021.Two studies were not found in Web of Science; thus citation tracking was performed in ResearchGate [40,47]. Of the 28 identified records, two were duplicates and 15 had emerged during our initial searches. The remaining 11 records were screened for titles and abstracts and one study required full-text analysis [52], but was excluded because PhA was not assessed. Two experts were also contacted, with three additional articles being suggested; since two of them had already been eliminated in the initial searches, only one new study [13] was included.

Ultimately, thirteen studies were included in our review [9,11,13,16,37,38,40,41,43,45,47,50,51]. Of these, only one had a pre-registered protocol [37], which we retrieved. Some studies provided a specific code for ethics registration, and the authors were contacted through email and, if available, by ResearchGate. Five authors provided protocols submitted to ethics committees [16,37,40,43,45]. These protocols were especially relevant for assessing risk of bias due to selective reporting. Figure 1 synthesizes the search and selection process.

### 3.2. Study Characteristics

The samples used in the studies included in this systematic review comprised a minimum of 12 [11] and a maximum of 273 athletes [41]. The total sum of samples from all studies was 1058 (65% male and 35% female), including practitioners of various sports (football, judo, volleyball, ice hockey, swimming, kendo and table tennis). The data are presented in Table 2.

The studies reported an average age ranging from 13.9 to 58.0 years, with the lowest age being 13 years [45] and the highest being 91 years [36], who, in this case, trains the elderly in his sample with master athletes. As for weight (sample size), only the volleyball group was included for the purposes of our review, considering the study by Di Vincenzo et al. [11]. Some studies aimed to verify the relationship between PhA (WB BIA or parts of the lower/upper limbs) and the strength of upper and lower limbs (e.g., in vertical jump, hand grip, isokinetic activities), whereas in some of the aforementioned studies, they carried out, in their methodologies, dissimilar BIA devices, analyzes between the control group and the intervention group, and also analyzed collectively or individually [16,37,43,45]. One study sought to verify the effects of age on body composition parameters, BIA data and performance in vertical jump or HGS and to better understand these effects during different stages of the menstrual cycle [39].

### 3.3. Risk of Bias in Studies

Table 3 describes the results of the risk of bias by using Cochrane’s RoBANS [19].

Of all 13 studies analyzed, none were considered as having overall low risk of bias; that is, 54% were denoted as unclear and 46% being at high risk of bias in at least one domain. For the selection of participants, all studies were considered at low risk of bias, while in the case of the study conducted by Di Vincenzo et al. [11], for the purposes of our analysis, we only considered the group of volleyball players. As for confounding variables, nine studies were considered to be at low risk of bias [9,11,13,37,38,40,43,50,51] and four studies at high risk of bias [16,41,43,47] for having differences between sex, age, competitive levels and sport practiced, the arbitrary division of analysis of groups or the division of analysis groups with sample numbers that were too different between groups, with no randomization or without even justification.

Regarding the measurement domain, ten studies were considered to be at a low risk of bias [11,13,16,37,38,40,41,45,50,51]. In contrast, two studies [43,47] were considered to be at a high risk of bias, because there were several concerns regarding endurance tests and the long jump was evaluated by a rudimentary method, respectively. As for the study conducted by Mala et al. [9], it is unclear because there was incomplete information provided for the measurements. As for the blinding evaluation of the results, all studies were considered to be at a low risk of bias, where, although there was no blinding, the procedures described for the tests indicated that it is unlikely the testers could have influenced the results, except for the study of Martins et al. [43], which was considered to be at high risk of bias, as there was no blinding that could have interfered with the standing long jump (SLJ) assessment.

In the incomplete outcome data domain, nine studies were considered to be at low risk of bias [9,11,13,37,38,40,43,50,51]. In contrast, three studies [16,41,45] were unclear due to insufficient information to assess, but it is possible that the base recruitment was larger than the reported group. Furthermore, the study conducted by Čerňanová et al. [47] was considered to be at high risk of bias, as data from only 21 of the 26 players declared to participate in the study were presented.

Finally, two studies were at high risk in selective outcome reporting [37,40], whereas the other eleven studies [9,11,13,16,38,41,43,45,47,50,51] were unclear as there was no pre-test protocol registered or ethics document available for comparison and evaluation.

### 3.4. Results of Individual Studies

The results of each study included in this research are presented in Table 4.

### 3.5. Data Synthesis

Considering the reduced number of studies and their heterogeneity, only one meta-analysis could be performed. The results of the meta-analysis that aimed to examine the relationship between PhA and lower limb strength (assessed by CMJ) are described in Figure 2.

Four studies were considered due to their similarity [38,43,50,51]. CMJ height was significantly associated with PhA (r = 0.691 [95% CI 0.249 to 0.895], Z = 2.797; *p* = 0.005). Tests for heterogeneity were identified as significant and high (*I*^2^ = 92.7%, Q = 41.3; *p* < 0.001). Thus, we observed a positive and significant correlation between PhA and CMJ. However, it was also noticed that one of the studies is a clear outlier, and as such, separate analyses were carried out (excluding one study at time) to examine whether a probable value of *I*^2^ could be smaller, as well as the amplitude of the CIs (Figure 3, Figure 4, Figure 5 and Figure 6).

The study conducted by Honorato et al. [51] (Figure 6) was removed for analysis because it presented a bias in the meta-analysis, with a large CI and not being significant in the relationship between PhA and CMJ. This study appeared to bias the general analysis, since it was a clear outlier in the analysis, as it was not statistically significant, since it had a very large CI.

### 3.6. Certainty of Evidence

Indirectness could be considered low for all outcomes, deriving from highly specific eligibility criteria. For the relationship between PhA and upper limb strength, the studies were few and heterogeneous, precluding a pooling of data (i.e., meta-analysis). In addition, there was a reduced number of participants (<800) and an unclear-to-high overall risk of bias. Therefore, currently the certainty of evidence should be deemed [53] very low for the relationship between these two outcomes.

Regarding the relationship between PhA and lower-limb strength, meta-analytical treatment was possible as four studies had sufficiently similar interventions and outcomes (i.e., CMJ) to pool their data. The reduced number of participants, unclear-to-high risk of bias in studies, impossibility of assessing risk of publication bias (<10 studies available) and high impact of statistical heterogeneity suggest very low confidence in the evidence, despite a clear direction of effects showing a positive association between PhA and lower-limb strength as assessed through CMJ.

## 4. Discussion

This systematic review reports a meta-analysis that evaluated the relationship between PhA (assessed by BIA devices) and lower- and upper-limb strength in athletes of both sexes in different sports. Of the 13 studies selected in this systematic review, a certain variety of studies that related PhA and strength were observed. Studies were found where the sample was composed of male and female athletes, between 13 and 91 years, as well as several sports (e.g., soccer, volleyball, judo, hockey, cycling) in several countries (Brazil, Portugal, Japan, Spain, Italy, Slovakia and Czech Republic). Albeit different brands of equipment to perform the BIA test were used in the studies, the equation used to calculate the PhA was always the same [(PhA = −arctangent (Xc/R) × (180/π)]. Moreover, 11 studies presented results that related PhA to strength of the upper and lower limbs, while only 2 studies did not show this correlation, and where a variety of physical tests were used to evaluate the lower limbs (CMJ, SLJ, 1RM, MYOTEST, 10 m and 30 m sprint times) and upper limbs (HGS and MYOTEST). All this plurality made us better understand the present theme has aroused interest in the international sports science community.

In the nine studies where the relationship between PhA and lower-limb strength was verified, eight of them presented positive correlations [13,37,38,43,45,47,50,51] and only 1 study [40] had no correlation. Moreover, five of the abovementioned studies had at least one domain with a high risk of bias, and the other four demonstrated at least one unclear domain, as in most domains of these studies there was a low risk of bias. Other strength tests for lower limbs were also used, such as 1RM (squat with barbell and leg press) and MYOTEST, and both showed positive correlations [13,47].

The four studies analyzed [40,43,50,51], which intended to verify the relationship between PhA and lower limb strength, revealed a great impact of statistical heterogeneity (*I*^2^ = 92.7%) and very wide confidence intervals; as such, everything will contribute to a GRADE in which the certainty of the evidence is very low. Moreover, one of the abovementioned studies [51] presented a bias in the meta-analysis, with a large CI and the relationship between PhA and CMJ not being significant.

The relationship between PhA and upper limb strength was verified in four studies, where three of them [9,11,16] presented these positive correlations (moderate and strong), and only one study [41] had no correlation. Probably because these last-mentioned authors reported that PhA was often associated with strength and physical fitness in adult and adolescent athletes, just as PhA was also associated with strength of manual prehension in healthy adult men. Additionally, this study was carried out in an age group with little variation in PhA; therefore, PhA could present a constant behavior in the regression models, and no significant difference in all the analyses used was found. Moreover, three of these studies [13,16,45] were considered to have at least one domain at high risk of bias and only one [11] was assessed in an unclear domain. However, in most domains of these studies, there was a low risk of bias.

## 5. Conclusions

In summary, the present research found that most studies showed a positive correlation between PhA and CMJ or HGS. The meta-analysis showed a relationship between PhA and lower-limb strength during CMJ. However, little is known regarding the upper limbs, as it was not possible to perform a meta-analysis, and for the lower limbs we performed the analysis with four studies and only with CMJ. Furthermore, GRADE showed very low certainty of evidence. In addition, we detected a high heterogeneity in meta-analysis results.

## 6. Other Information

This systematic review was registered on the Open Science Framework (OSF) on 30 November 2021, i.e., one day before the initial searches were performed. Link to project: https://osf.io/pmhgq/. Link to registration: https://osf.io/f5vxy.

## Figures and Tables

**Figure 1 sports-11-00107-f001:**
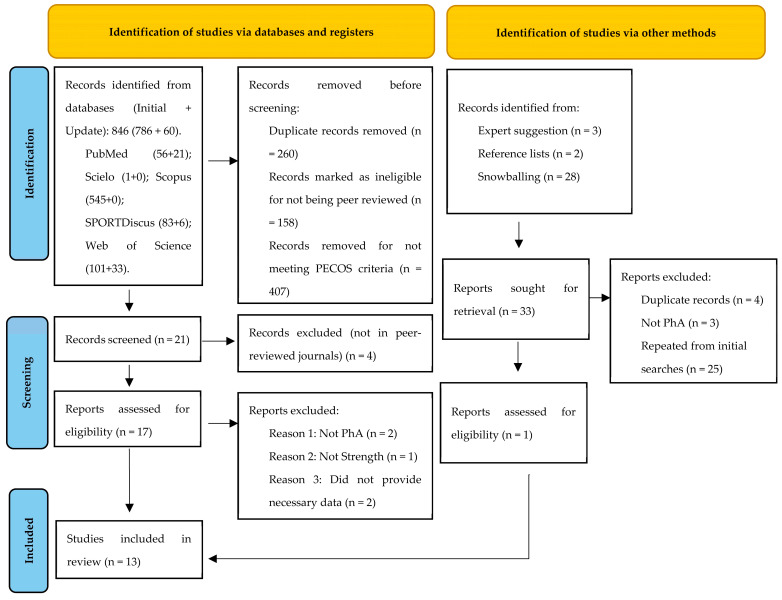
PRISMA flow diagram highlighting the selection process for the studies included in the systematic review.

**Figure 2 sports-11-00107-f002:**
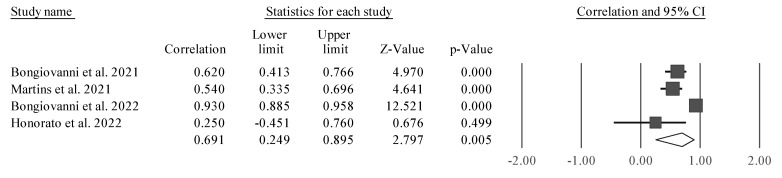
Forest plot outlining the association between CMJ height and PhA. Values shown are effect sizes (Hedges’s g) with 95% confidence intervals (CI). The size of the plotted squares reflects the statistical weight of the study. Black diamond: overall results [38,43,50,51].

**Figure 3 sports-11-00107-f003:**
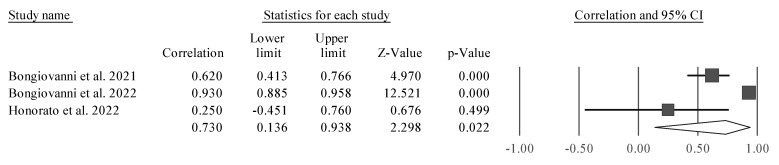
Forest plot describing the association between CMJ height and PhA after removing the study by Martins et al. (2021) [43]. Values shown are effect sizes (Hedges’s g) with 95% confidence intervals (CI). The size of the plotted squares reflects the statistical weight of the study. Black diamond: overall results [38,50,51].

**Figure 4 sports-11-00107-f004:**
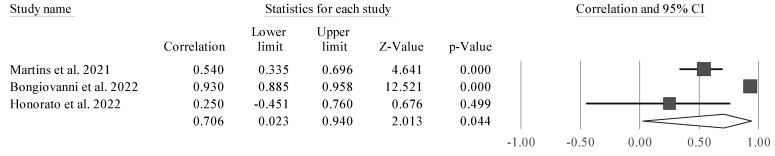
Forest plot describing the association between CMJ height and PhA after removing the study by Bongiovanni et al. (2021) [38]. Values shown are effect sizes (Hedges’s g) with 95% confidence intervals (CI). The size of the plotted squares reflects the statistical weight of the study. Black diamond: overall results [43,50,51].

**Figure 5 sports-11-00107-f005:**
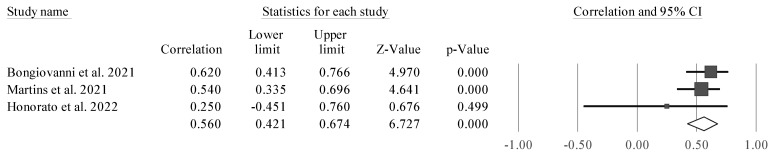
Forest plot describing the association between CMJ height and PhA after removing the study by Bongiovanni et al. (2022) [50]. Values shown are effect sizes (Hedges’s g) with 95% confidence intervals (CI). The size of the plotted squares reflects the statistical weight of the study. Black diamond: overall results [38,43,51].

**Figure 6 sports-11-00107-f006:**
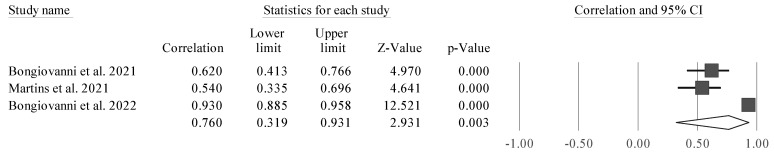
Forest plot describing the association between CMJ height and PhA after removing the study by Honorato et al. (2022) [51]. Values shown are effect sizes (Hedges’s g) with 95% confidence intervals (CI). The size of the plotted squares reflects the statistical weight of the study. Black diamond: overall results [38,43,50].

**Table 1 sports-11-00107-t001:** Full search strategies for each database.

Database	Search Strategy
PubMed	(((bioimpedance OR “bioelectrical impedance”) AND (“phase angle”)) AND (strength OR force OR power OR potenc* OR muscle OR muscul*) AND (athlet* OR sport*))
Scielo	(bioimpedance OR “bioelectrical impedance”) AND (“phase angle”) AND (stregth OR force OR power OR potenc* OR muscle OR muscul*) AND (athlet* OR sport*)
Scopus	(ALL (bioimpedance OR “bioelectrical impedance”) AND ALL (“phase angle”) AND ALL (strength OR force OR power OR potenc* OR muscle OR muscul*) AND ALL (athlet* OR sport*))
SPORTDiscus	TX (bioimpedance OR “bioelectrical impedance”) AND TX “phase angle” AND TX (strength OR force OR power OR potenc* OR muscle OR muscul*) AND TX (athlet* OR sport*)
Web of Science	ALL FIELDS: (bioimpedance OR “bioelectrical impedance”) AND ALL FIELDS: (“phase angle”) AND ALL FIELDS: (strength OR force OR power OR potenc* OR muscle OR muscul*) AND ALL FIELDS: (athlete* OR sport*)

**Table 2 sports-11-00107-t002:** Characteristics of the included studies.

Study Country Where the Research Took Place	*n* Sex	Age	Competitive Level	Physical Outcomes	Description of Interventions
Alvero-Cruz et al. [37] Spain	Total: 256 Male: 162 Female: 91	58.0 ± 12.0	Master athletes	PhA CMJ	A brief questionnaire-guided interview was performed with the participants to assess information on athletic specialization, training habits and medical conditions. All participants were subjected to BIA (InBody S10, Seoul, Republic of Korea) with a segmental multifrequency approach. The test was performed with a Leonardo ground reaction force platform (Novotec Medical, Pforzheim, Germany) with the integrated software in its 4.4b01.35 version (research addition).
Bongiovanni et al. [50] Italy	Male: 15	28.7 ± 5.0	Elite	PhA CMJ	To assess PhA, a BIA 101 Biva Pro (Akern, Florence, Italy) was used. Whole-body and lower hemisoma PhA were obtained with a phase-sensitive 50 kHz BIA and leg lean soft tissue. It was estimated using a specific bioimpedance-based equation developed for athletes. Vertical jump performance was assessed using CMJ.
Campa et al. [40] Italy	Female: 20	23.8 ± 3.4	Elite	PhA CMJ	The procedures were synchronized individually between all participants so as to have a familiarization session before the first early follicular phase and 4 testing assessments. In particular, the testing assessments were performed on the second day of each early follicular phase and 14 days later, when the participants were in their ovulatory phase. To assess body composition, BIA (BIA 101 Anniversary; Akern, Florence, Italy) was performed, and BIVA procedures were applied. To assess performance, CMJ and 20-m sprint tests were used.
Cattem et al. [41] Brazil	Total: 273 Male: 161 Female: 112	12.9 ± 0.9	Beginners	PhA HGS	The adolescent students were classified as athletes according to the Sports Dietitians Australia Position Statement. To evaluate PhA, BIA measurements were always performed in the morning, using a tetrapolar analyzer RJL (Quantum 101; Systems, Clinton Township, MI, USA), at a single frequency of 50 kHz. Participants were in the supine position with a leg opening distant from the median line of the body and the upper limbs distant from the trunk. HGS was assessed with a hand JAMAR-dynamometer (Asimow Engineering Co., Los Angeles, CA, USA) in both hands alternately, three times, and the mean value was recorded to obtain a single value of HGS.
Čerňanová et al. [47] Slovakia	Male: 21	G1: 15.18± 0.75 G2: 17.14± 0.9	Competitive	PhA CMJ HGS	Ice hockey players were divided into two training groups, one group with collective training (*n* = 18; 13 completed the study) and one group with individual training (*n* = 8). Physical performance parameters included upper and lower limb power, force and velocity. Body composition analysis was determined by BIA device (BIA 101-Akern, Florence, Italy) and BODYGRAM software (version 1.3 for Windows) and MYOTEST PRO diagnosed the force and speed–force components of the upper and lower limbs.
Di Vicenzo et al. [11] Italy	Female: 12	23.8 ± 3.6	Competitive	PhA HGS	Participants included elite female volleyball players on a team of the Italian Serie B League and twenty-two young women with similar characteristics who served as the control group. They trained six days/week for about 4 h/day. Control women were selected from among Federico II University students. Assessment of PhA was carried out using a tetrapolar unifrequency BIA device (BIA 101 Anniversary Akern, Florence, Italy) at a frequency of 50 kHz. Upper-limb muscle strength was based on HGS, assessed using a Jamar handgrip dynamometer (Asimow Engineering, Santa Fé Springs, CA, USA). Maximum relative lower-limb power and maximum average power of the lower limbs was assessed in the form of a CMJ, using the Leonardo Mechanograph Ground Reaction Force Plate (GRFP; Novotec Medical GmbH, Pforzheim, Germany).
Hetherington-Rauth et al. [16] Portugal	Total: 117 Male: 57 Female: 60	20.9 ± 3.5 21.1 ± 4.1	Competitive	PhA CMJ HGS	Muscle performance was assessed in athletes from several sports, which consisted of a measure of upper-body strength and lower-body power. PhA of the upper and lower limbs were correspondingly measured. WB assessment of PhA was carried out using a tetrapolar unifrequency BIA device (BIA 101 Anniversary Akern/RJL Systems; Florence, Italy) at a frequency of 50 kHz. Upper-limb muscle strength was based on HGS assessed using a Jamar handgrip dynamometer (Asimow Engineering, Santa Fé Springs, CA, USA). Maximum relative lower-limb power and maximum average power of the lower limbs was assessed in the form of a CMJ using the Leonardo Mechanograph Ground Reaction Force Plate (GRFP; Novotec Medical GmbH, Pforzheim, Germany).
Mala et al. [9] Czech Republic	Total: 59 Male: 39 Female: 20	12.08 ± 1.47	Cadet and junior teams	PhA HGS	Judo athletes gripped a dynamometer with maximal effort in a sitting position with full extension of the elbow in two trials for each limb and with a rest interval lasting 60 secs between the trials. To assess whole-body bio-impedance, we used a Tanita MC-980MA multi-frequency bio-impedance analyser (Tanita Corporation, Japan). Only the best performance in the trial was processed in the subsequent analysis. The participants’ writing hand was used as the preferred upper limb.
Martins et al. [43] Brazil	Male: 62	15.0 ± 1.4		PhA CMJ	Male youth soccer players were evaluated for PhA and physical performance attributes, the evaluation consisting of standing long jump (SLJ), IER capacity, sprinting speed and repeated sprint ability (RSA). The first week of testing included only body composition assessments by means of BIA. To assess PhA, a BIA octopolar multi-frequency equipment (Biospace, Los Angeles, CA, USA) was used. During the second week, two days of the training microcycle were dedicated to the application of the following physical tests to all players: (i) on the first day, standing long jump (SLJ) and Carminatti’s test (T-CAR) and (ii) on the second day, straight sprint test and RSA protocol.
Obayashi et al. [45] Japan	Total: 170 Male: 110 Female: 60	13.9 ± 1.6	Competitive	PhA CMJ	All participants’ height and weight were assessed and entered into the device (In Body S10 Body Water Analyzer; InBody Co., Seoul, Republic of Korea) before starting BIA. The measurements were taken in the supine position with no limbs in contact with each other. CMJ height and squat jump (SJ) height were measured as jump parameters.
Bongiovanni et al. [38] Italy	Male: 16	14.3 ± 1.0	Elite	PhA CMJ	An observational study design was adopted to assess the contribution of whole-body and regional raw bioelectrical BIA parameters on performance in a group of U14 elite soccer players. Athletes underwent whole-body and regional BIA analysis in a fasting state. All players were requested to abstain from using dietary supplements, from drinking caffeinated drinks and from exercising at moderate-to-high intensity (except during the tests included in the experimental design) before (within 48 h) and on the day of the study. To assess PhA, a BIA 101 Biva Pro (Akern, Florence, Italy) was used. For CMJ, the Optojump Next System (Microgate, Bolzano, Italy) was used to indirectly record vertical-jump height for each participant.
Honorato et al. [51] Brazil	Male: 10	30.0 ± 4.5	Elite	PhA CMJ	The present research was a quasi-experimental study delineated to assess the effects of a six-week pre-season period on BIA-derived parameters, body composition components, power, and aerobic abilities in professional soccer players. The Quantum V Segmental BIA^®^ 152 bioimpedance device (RJL Systems^®^) at a fixed frequency of 50 kHz was used for whole-body 153 and regional BIA measurements.
Cesanelli et al. [13] Italy	Male: 30	26.33± 3.61	Amateurs and sub-elite	PhA 1RM	This was a longitudinal study in which data were acquired at a one-year strength and conditioning training program of well-trained cyclists. Pre- and post-values of performance indicators, body mass composition and strength were compared to assess the impacts of the one-year strength program. BIA was performed to evaluate body composition using a BIA Akern 101 device (Akern, Florence, Italy).

**Table 3 sports-11-00107-t003:** Risk of Bias (RoBANS).

Study	Selection of Participants	Confounding Variables	Measurement	Blinding Outcome Assessment	Incomplete Outcome Data	Selective Outcome Reporting
Alvero-Cruz et al. [37]	Low risk	Low risk	Low risk	Low risk	Low risk	High risk
Bongiovanni et al. [50]	Low risk	Low risk	Low risk	Low risk	Low risk	Unclear
Campa et al. [40]	Low risk	Low risk	Low risk	Low risk	Low risk	High risk
Cattem et al. [41]	Low risk	High risk	Low risk	Low risk	Unclear	Unclear
Čerňanová et al. [47]	Low risk	High risk	High risk	Low risk	High risk	Unclear
Di Vincenzo et al. [11]	Low risk	Low risk	Low risk	Low risk	Low risk	Unclear
Hetherington-Rauth et al. [16]	Low risk	High risk	Low risk	Low risk	Unclear	Unclear
Mala et al. [9]	Low risk	Low risk	Unclear	Low risk	Low risk	Unclear
Martins et al. [43]	Low risk	Low risk	High risk	High risk	Low risk	Unclear
Obayashi et al. [45]	Low risk	High risk	Low risk	Low risk	Unclear	Unclear
Bongiovanni et al. [38]	Low risk	Low risk	Low risk	Low risk	Low risk	Unclear
Honorato et al. [51]	Low risk	Low risk	Low risk	Low risk	Low risk	Unclear
Cesanelli et al. [13]	Low risk	Low risk	Low risk	Low risk	Low risk	Unclear

**Table 4 sports-11-00107-t004:** Results of individual studies.

	Authors of the Study	Type of Study	Aim	Main Results and Findings
**PhA and Lower Limb Strength**	Alvero-Cruz et al. [37]	Experimental study composed of 256 master athletes; of these 240 athletes were between 35 and 91 (58.0 ± 12.0 years)	To investigate whether age-related effects in body composition could explain the age-related decline in vertical jumping performance of master athletes.	The results obtained demonstrated moderate PhA correlation in males (r = −0.32 to 0.67, *p* < 0.0001) and larger correlation coefficients in females (r = −0.470 to 0.820, *p* < 0.0001) with CMJ.
	Bongiovanni et al. [50]	A pilot longitudinal study design composed of fifteen elite soccer players (28.7 ± 5.0 years) from the first Italian division (Serie A).	To verify the association between changes in lower PhA and CMJ in elite soccer players.	The results were PhA Pre—7.9° ± 0.5; Post: 8.0° ± 0.4; 95% CI: −0.35, 0.09; t: −1.2 and CMJ Pre—49.5 cm ± 7.8; Post: 50.1± 4.8; 95% IC: −1.63, −0.20; t: −2.7. The major findings were that changes in lower PhA were more strongly related with changes in CMJ (r^2^ = 0.617, *p* = 0.001) than changes in WB PhA (r^2^ = 0.270, *p* = 0.047).
	Bongiovanni et al. [38]	Experimental study using sixteen male elite soccer players (14.3 ± 1.0 years) from the same club competing in the Italian first division.	To examine the association between regional (UPhA and LPhA) and total (WB PhA) PhA in sprinting and jumping performance in soccer players.	The results showed monitoring regional PhA was more informative than total PhA in sprint and vertical-jump performance in young elite soccer players. This study showed a moderate correlation between PhA and CMJ (r = 0.680; *p* < 0.001).
	Campa et al. [40]	Experimental study using a total of 20 female soccer players (23.8 ± 3.4 years).	To analyze the fluctuations in body composition and bioelectrical parameters assessed by BIA and CMJ in jumping and running abilities and flexibility of elite soccer players.	The results of PhA (6.7° ± 0.6°) in elite soccer players showed that CMJ (29.4 cm ± 4.1) and sprinting capacity were not affected, whereas flexibility decreased during the early follicular phases. Also showed a moderate correlation between PhA and CMJ (r = 0.568; *p* = 0.009).
	Honorato et al. [51]	Experimental study that evaluated the effects of a six-week pre-season period on whole-body and regional BIA derived parameters in professional soccer players.	To assess body composition component and neuromuscular and aerobic performance changes in response to the pre-season training period.	The results suggested it is possible to infer that the regional BIA-derived parameters, more specifically the hamstrings PhA (M1 = 10.9° ± 2.3; M2 = 10.8° ± 2.3; M3 = 11.7° ± 2.4), were augmented after six weeks of pre-season training in athletes. The same was not observed for the WB PhA. This study showed a weak correlation between PhA and CMJ both for the pre- and post-test time (pre—0.180 and post—0.250).
	Martins et al. [43]	Cross-sectorial study that evaluated sixty-two adolescent male players (15.0 ± 1.4 years) from two professional soccer academies of the Brazilian National League (14 were Under-13, 25 were Under-15, and 23 were Under-17).	To verify the association between PhA and components of physical performance in male youth soccer players.	The results verified PhA (U13 = 6.1° ± 0.6; U15 = 5.2° ± 0.4; U17 = 6.2°± 0.4; F = 26.8; *p* ≤ 0.01) is associated with 10 m and 30 m sprint times and RSA performance in young male soccer players, regardless of age-related variability and body composition measures. The multiple regression analysis outputs showed that PhA remained inversely related to test 10 m (β = −0.379; *p* = 0.012) and 30 m ( β = −0.438; *p* < 0.001) sprint times, while the association with standing long jump (SLJ) performance were statistically non-significant.
	Obayashi et al. [45]	Experimental study that included 170 adolescent athletes (13.9 ± 1.6 years) who underwent a sports medical check-up, including body composition and physical performance tests.	To investigate the association between PhA and physical performance in adolescent athletes.	The results were: PhA (6.0° ± 0.7; W = 0.98; *p* = 0.04) and CMJ (28.2 cm ± 5.9; W = 0.98; *p* < 0.01). They concluded that WB PhA was correlated with upper- and lower-limb muscle strength and jump performance in adolescent athletes.
	Cesanelli et al. [13]	Experimental study that analyzed thirty well-trained male (26.33 ± 3.61 years) competitive cyclists (amateurs and sub-elite categories).	To investigate the effect of a combined one-year strength and conditioning training program on performance indicators and body composition and to determine the possible relationships between these variables	The results (PhA: 6.89° ± 0.43 and 6.97° ± 0.46; 1RMtot (kg) 62.85 ± 28.0 and 105.42 ± 47.4) of this study indicated beneficial impacts from one-year combined strength and conditioning training on cycling performance indicators and demonstrated correlation between the performance indicators (athletes’ threshold power, body composition and strength), suggesting the possible existence of different adaptation zones. Also showed a very weak correlation between PhA and 1RM (Leg Press) (r = 0.160; *p* = 0.001).
**PhA and Upper Limb Strength**	Cattem et al. [41]	Cross-sectorial study in which 273 Brazilian healthy adolescents (161 males, 12.9 ± 0.9 years) engaged in different sports and were evaluated.	To analyze the efficiency of BIA device (PhA), considering chronological age and HGS in male adolescent athletes.	Although PhA is often associated with strength and physical fitness in adult athletes and adolescent athletes, PhA was also associated with HGS in healthy adult men (β = 0.058; *p* = 0.114).
	Di Vicenzo et al. [11]	Experimental study of twelve volleyball players (23.8 ± 3.6 years) and 22 non-athletic females, who served as a control group (23.6 ± 2.0 years).	To evaluate body composition and segmental PhA for both WB and segmental limbs in twelve elite female volleyball players compared to a group of twenty-two non-athletic controls and to investigate the possible relations between PhA and muscular strength assessed by HGS.	The results obtained for the volleyball player group was WB PhA (6.8° ± 0.43; *p* < 0.001); HGS (25.4 kg ± 4.3; *p* = 0.358). They conclude there is a clear relationship between HGS and PhA in athletes (r = 0.696, *p* = 0.012).
	Mala et al. [9]	Experimental study of 59 judo athletes (39 boys and 20 girls), all members of the Czech cadet and junior teams.	To investigate gender differences in body composition, muscle strength in upper limbs, upper- and lower- limb morphology, and upper-limb strength among adolescent judo athletes.	In the non-dominant upper limb, we detected a significant correlation between PhA and the level of muscle strength (boys: r = 0.64, *p* < 0.01, girls: r = 0.61, *p* < 0.01).
**PhA and Lower and Upper Limb Strength**	Hetherington-Rauth et al. [16]	Experimental study of 117 adult athletes recruited from different national clubs in Lisbon, Portugal.	To examine the associations of muscle strength and power with PhA in national- and international-level athletes from different sports and to assess if these associations were independent of lean soft tissue (LST).	In the results obtained, (PhA—Female = 6.8° ± 0.6, Male = 7.9° ± 0.7, All = 7.3° ± 0.8; HGS—Female = 33.4 kg ± 5.0; Male = 49.8 kg ± 7.9; All = 41.4 kg ± 10.5), WB PhA was related to both upper (β = 0.86) body strength and lower (β = 0.81) body power.
	Čerňanová et al. [47]	Experimental study of 21 young male ice hockey players (15–18 years).	To evaluate potential differences in body composition and physical performance between ice hockey players with different training approaches during preseason preparation.	The average of the results (PhA: Colletive = 7.84° ± 0.31; Individual = 8.84° ± 0.65, *p* = 0.043; Force of lower limbs: Collective = 1909.70 N ± 175.89; Individual = 2710.77 N ± 261.33, *p* < 0.001) showed that in the athletes who received the collective training approach, both for upper limb power (*p* = 0.809) and for lower limbs (*p* = 0.888), there was a correlation with the PhA variable. As for the athletes who received training using the individual approach, there was a correlation with power in the upper limbs (*p* = 0.911), with no such correlation for the lower limbs in relation to PhA. The average correlation in the two training approaches (Collective and Individual) between PhA and strength performance parameters were upper limb strength (N): r = 0.767, *p* = 0.001 and lower limb strength (N): r = 0.726, *p* = 0.002.

## Data Availability

Our data were made available with the submission.

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
