# Peer review of "Relationship between Bioelectrical Impedance Phase Angle and Upper and Lower Limb Muscle Strength in Athletes from Several Sports: A Systematic Review with Meta-Analysis"

_sports, 2023, doi:10.3390/sports11050107_

Round 1

Reviewer 1 Report

The manuscript is interesting and well written. I have some minor suggestions for authors. If possible, the tables should be reviewed trying to make them easier to interpret. Also the resolution of figure 1 should be increased (Prisma flow diagram).

Author Response

Dear Reviewer,

As proposed, we simplify the description of the tables and improved the resolution of figure 1 (all the changes are in yellow).

Many thanks for your consideration.

Reviewer 2 Report

Dear authors,

The study is relevant and current. I suggest that the analyzer used in each study be highlighted, whether it was single, multi, or BIS, highlighting the possible lack of agreement between PhA values obtained by different devices.

Page 2 line 100 – “No reviews were reported specifically the relationships between PhA with strength in athletes (lower and upper limbs), this demonstrates there is a gap in the literature on this domain…” What is the hypothesis for this fact to happen?

Author Response

Dear Reviewer,

As proposed, we have changed the manuscript accordingly to your suggestions (in yellow).

Point 1: The study is relevant and current. I suggest that the analyzer used in each study be highlighted, whether it was single, multi, or BIS, highlighting the possible lack of agreement between PhA values obtained by different devices.

Response 1: we added at the line 251 "dissimilar BIA devices" and line 358 "Albeit different brands of equipment to perform the BIA test were used in the studies, the equation used to calculate the PhA was always the same [(PhA= −arctangent (Xc/R) * (180/π)]." Moreover, we included all brands of the equipment used by the researchers in Table 2.

Page 2 line 100 – “No reviews were reported specifically the relationships between PhA with strength in athletes (lower and upper limbs), this demonstrates there is a gap in the literature on this domain…” What is the hypothesis for this fact to happen?

Response 2: We rephrased the sentence to "No reviews reported specifically the relationships between PhA with strength in athletes (lower and upper limbs), perhaps denoting the interest in the field still has not achieved critical mass. This demonstrates there is a gap in the literature on this domain, in which we expected that this research may contribute to the construction of knowledge and state of the art for the sport sciences."

Many thanks for your considerations.

Reviewer 3 Report

Dear Author,

Thank you for your effort and time on your manuscript. I have provided the necessary recommendations for your paper. It is important to congratulate the authors for the work carried out, despite the limitations of the research.

This systematic review with a meta-analysis aimed to examine the existence of relationship between PhA and the strength of lower and upper limbs of both sexes in athletes of different sports.

This study appears to be novel and the methodology is clear.

Specific comments are provided below:

TITLE

The Title is good and clear.

INTRODUCTION

The introduction is clear and leads to the rationale of the study. The importance of the study is very well mentioned at the end of the section.

Line 90: "lean soft tissue" please correct it

MATERIALS AND METHODS / RESULTS

The methodology and results are described and presented very well. All information are presented with details and the tables and Figures are clear and helpful for the reader.

Table 1: Correct the parenthesis in the second line

Figure 1: Correct the red line words "Scielo, SPORTDiscus, PhA"

Table 2 and Table 3: Correct the formation of the cells

Lines 254 - 256, 259, 264 - 265, 269, 272, 274, 277, 374 - 375: replace the word "judged" with another word

DISCUSSION AND CONCLUSION

These sections are very well written. Well done!

The quality of the English Language is fine

Author Response

Dear Reviewer,

As proposed, we updated the manuscript accordingly to your suggestions, namely:

Point 1: Line 90: "lean soft tissue" please correct it.

Response 1: Corrected.

Point 2: Table 1: Correct the parenthesis in the second line

Response 2: Corrected.

Point 3: Figure 1: Correct the red line words "Scielo, SPORTDiscus, PhA"

Response 3: Corrected.

Point 4: Table 2 and Table 3: Correct the formation of the cells

Response 4: Corrected.

Point 5: Lines 254 - 256, 259, 264 - 265, 269, 272, 274, 277, 374 - 375: replace the word "judged" with another word

Response 5: You changed to "considered" and "assessed".

Many thanks for your consideration.
